# Microbial Gene Ontology informed deep neural network for microbe functionality discovery in human diseases

Yunjie Liu⊙*, Yao-zhong Zhang⊙, Seiya Imoto

Institute of Medical Science, The University of Tokyo, Tokyo, Japan

* liuyj@hgc.jp

## Abstract

The human microbiome plays a crucial role in human health and is associated with a number of human diseases. Determining microbiome functional roles in human diseases remains a biological challenge due to the high dimensionality of metagenome gene features. However, existing models were limited in providing biological interpretability, where the functional role of microbes in human diseases is unexplored. Here we propose to utilize a neural network-based model incorporating Gene Ontology (GO) relationship network to discover the microbe functionality in human diseases. We use four benchmark datasets, including diabetes, liver cirrhosis, inflammatory bowel disease, and colorectal cancer, to explore the microbe functionality in the human diseases. Our model discovered and visualized the novel candidates' important microbiome genes and their functions by calculating the important score of each gene and GO term in the network. Furthermore, we demonstrate that our model achieves a competitive performance in predicting the disease by comparison with other non-Gene Ontology informed models. The discovered candidates' important microbiome genes and their functions provide novel insights into microbe functional contribution.

## Introduction

With the development of metagenome sequencing technologies, a large number of studies were designed to find the association between the human microbiome and human disease [1–4]. The human microbiome has been shown to play an important role in diseases such as type II diabetes (T2D), liver cirrhosis, and obesity. Therefore, discovering the human microbiome's roles in human diseases will guide the researchers to explain these diseases from a metagenome aspect. However, describing the microbe's roles in human diseases remains a biological challenge due to the complexity of discovering and summarizing the microbe's roles with many microbes.

To address the problem, various machine learning and deep learning models were designed [5–9]. These models extracted different microbiome features and evaluated the significance of these features by predicting the diseases. LaPierre et al. [5] compared the performance of different machine learning and deep learning models in predicting human diseases using different metagenome features. They extracted the taxa abundance feature using MetaPhlAn2 [10]

**Data Availability Statement:** GO informed neural network is available from the GitHub repository (https://github.com/YunjieLiu-HGC/GO_NN). The datasets generated and analysed during the current study are publicly available from the Science Data

Bank repository (https://doi.org/10.57760/sciencedb.01684).

**Funding:** The author(s) received no specific funding for this work.

and $k$-mer abundance feature using Jellyfish [11] in their experiments and predicted the diseases using these features separately. They have shown the potential of deep learning models in disease predicting tasks. However, there are some limitations to these methods. On the one hand, the taxa abundance feature gives limited functional information and hampers people from understanding how the microbes affect the diseases. On the other hand, using $k$-mer abundance feature or deep learning models has limited biological interpretability. It is difficult to understand the mechanisms underlying the prediction results.

Instead of using the taxa abundance feature and $k$-mer abundance feature, functionality feature such as KEGG provides the functional aspect to explain the microbes' role in human diseases [12]. Traditionally, statistics-based models were used to identify the disease-associated function features [1, 13, 14]. These statistics-based models identify the disease-associated functions that significantly differ between case and control groups. However, these models were typically based on the linear or independent assumption. These models will not detect features that have a complex relationship with diseases.

Recently, a novel interpretable deep learning model named P-NET was developed to predict treatment resistance in prostate cancer patients using the biologically informed hierarchical structure [15]. They demonstrated that P-NET could predict cancer state using molecular data with a performance superior to other machine learning models. However, whether a microbe functionality-informed hierarchical structure can effectively predict the diseases is still unknown. Furthermore, solving the problem is challenging due to the high dimensionality of metagenome genes and functionality features.

Another study named ParsVNN was designed to discover the cancer-specific, and drug-sensitive genes [16]. ParsVNN used GO hierarchical structure in building the visible neural network and pruned the edges in the network to remove the redundant features. However, the network is not applicable in handling metagenome data due to the high dimensionality of metagenome genes and functionality features. The network will be constructed with too many parameters before pruning the edges.

To address the above problems, we proposed a novel interpretable deep learning model utilizing GO hierarchical structure, which describes genes in molecular function, biological process, and cellular component [17, 18], to interpret the microbiome functional roles in different human diseases. We performed our model on diabetes, liver cirrhosis, inflammatory bowel disease, and colorectal cancer, which showed the model's effectiveness. Furthermore, our model discovered the novel candidates' important microbiome genes and their functions by calculating the important score of each gene and GO term in the network in both diabetes and liver cirrhosis.

## Materials and methods

### Data preparation

The workflow of the pipeline is shown in Fig 1. Raw metagenome sequencing data obtained in previous studies were used as input in the pipeline. AfterQC is used to perform quality control, including filtering low-quality reads and trimming adapters [19]. Human genomes are removed using Bowtie2, where the reference genome is Genome Reference Consortium Human Build 38 (GRCh38) [20]. Reads after quality control are aligned to the Unified Human Gastrointestinal Genome (UHGG) collection [21] by bowtie2 to find the comprehensive gene representation. The quantification of genes is performed by using Salmon [22]. Transcript Per Million (TPM) is used as the gene quantification profile, which is used as the GO network-informed deep neural network (DNN) input. To obtain the GO annotation information, we first obtain the protein annotation from each gene provided by the UHGG and find out the

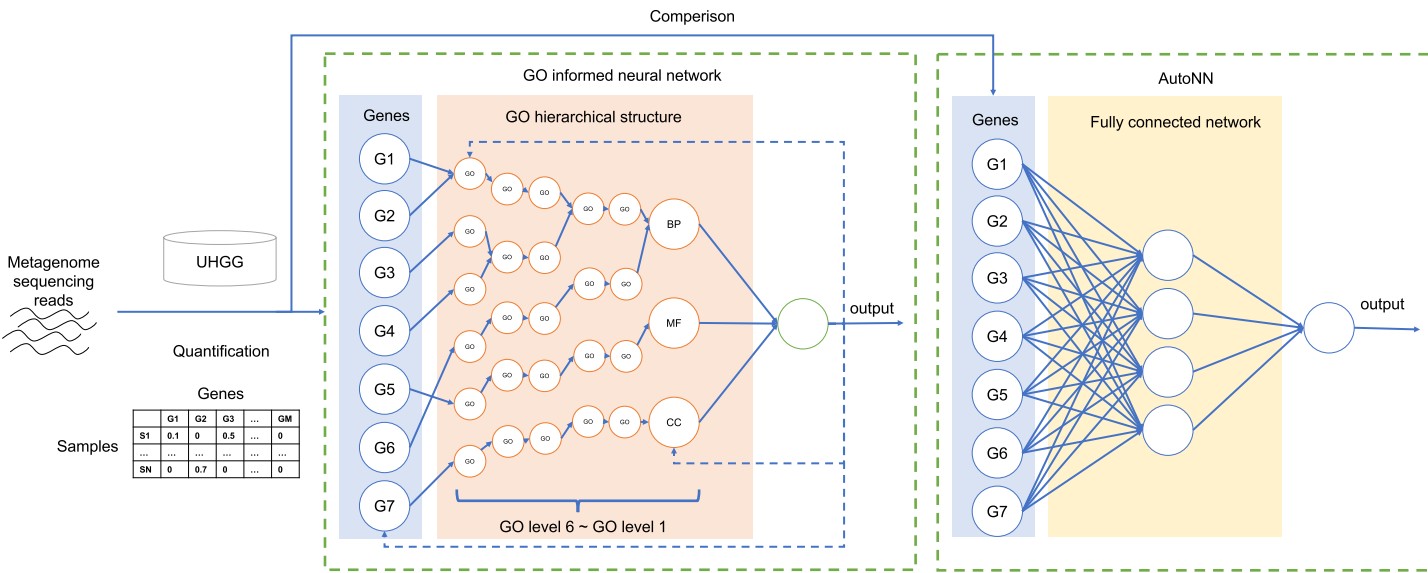

**Fig 1. The workflow of the pipeline.** Metagenome sequencing reads after quality control are aligned to the UHGG collection. After alignment, TPM is used as the gene quantification profile, which is used as the input of the GO-informed neural network and non-GO-informed neural network (AutoNN). In the GO-informed neural network, the solid arrow in the network is determined by Gene functional annotation information and GO hierarchical structure. The dashed arrows show the direction of calculating the importance score. The network output the disease prediction result and important genes and GO terms candidates. In AutoNN, the network is fully connected. Abbreviations: BP (biological process), MF (molecular function), and CC (cellular component).

gene-protein mapping. Then, we search the Gene Ontology annotation of each protein from the Uniprot database [23] to find the out the protein-GO mapping. Finally, by merging the gene-protein mapping and protein-GO mapping, we obtain the GO annotation for each UHGG gene.

In this study, we prepare metagenome sequencing data for four diseases shown in Table 1. We select commonly existing genes from the UHGG gene catalog to reduce the feature dimension. We prepare a gene set where the genes exist in more than 1% of UHGG samples as commonly existing genes. The number of genes selected from the UHGG gene catalog is 22,927. To compare our model and other machine learning models, we filtered out the non-GO annotated genes and obtained the final selected gene set with 8,010 genes.

## GO network informed DNN construction

GO describes genes in three aspects: molecular function, biological process, and cellular component, which are also three GO terms in the GO hierarchical structure. These three GO terms represent the roots of the ontologies, respectively. In our network, each node represents a GO term. There are different relations between GO terms where we choose the main relation: *is a,*

**Table 1. Summary of the datasets covered in this study.**

| Dataset | Case samples number | Control samples number | Citation |
|---|---|---|---|
| T2D | 187 | 172 | [1] |
| Liver cirrhosis | 162 | 145 | [2] |
| Inflammatory bowel disease | 163 | 56 | [24] |
| Colorectal cancer | 53 | 88 | [25] |

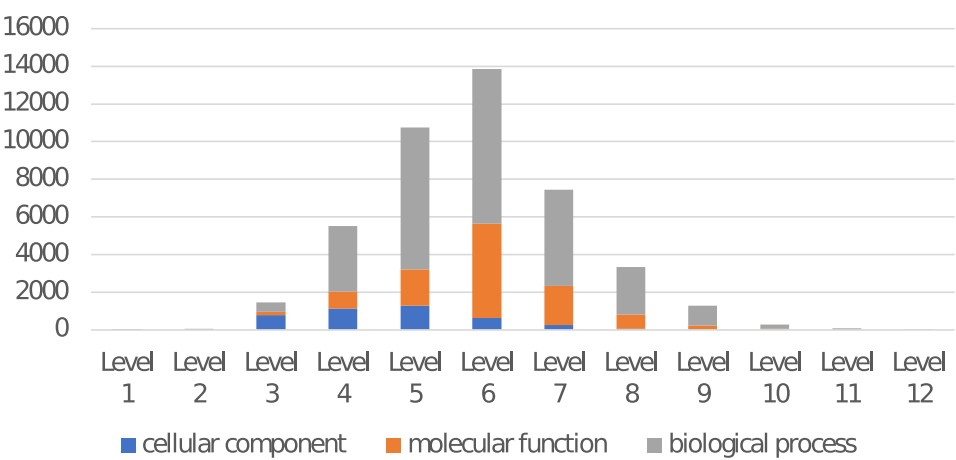

**Fig 2. GO terms distribution in each level.**

*part of, has part, regulates (including positively regulates and negatively regulates)* as edges in our research. We define the root nodes as level 1, children nodes that directly relate to the root nodes as level 2, and so on. The distribution of GO terms in each level is shown in Fig 2. The first six levels are used in constructing the neural network. Metagenome genes are annotated to level 6 by following rules: genes have annotation GO terms in level 1 to level 5 are not connected with level 6 GO terms; genes have annotation GO terms in level 6 to level 12 are connected with level 6 GO terms, where higher level GO terms are mapped to their ancestor GO terms in level 6.

In the whole neural network, the input layer $x_{gene}$ represents the metagenome genes quantification profile. The second layer represents the level 6 GO terms nodes. The output of the input layer is calculated as $y = f[(M * W)^T x_{gene} + \mathbf{b}]$, where $M$ is the mask matrix, $W$ is the weights matrix, $\mathbf{b}$ is the bias vector, $*$ is Hadamard product, and the activation function $f$ is $f = \tanh = (e^{2x} - 1)/(e^{2x} + 1)$. The mask matrix from genes to level 6 GO terms nodes is defined as a binary matrix $M \in \{0, 1\}^{D_x \cdot D_y}$ where $D_x$ is gene number and $D_y$ is the level 6 GO terms number. When the gene $i$ is annotated to GO terms $j$, $M[i, j] = 1$ ($1 \leq i \leq D_x$, $1 \leq j \leq D_y$); otherwise, $M[i, j] = 0$. The Hadamard product $*$ product each element of mask matrix $M$ and weight matrix $W$ to zero out all the connections that do not exist in the annotation. The second layer to the seventh layer represent the GO network where the output of each layer is calculated as $y = f[(M * W)^T x_{layer\_i} + \mathbf{b}]$ ($i = 6, 5, 4, 3, 2, 1$). The mask matrix zeros out the not connected GO terms in the network. We add a predictive layer with sigmoid activation $\sigma = 1/(1 + e^{-x})$ after each hidden layer to calculate the final prediction by taking the average of all the predictive elements in the network. In the whole neural network, $M$ is the mask matrix dependent on the GO annotation of the genes and GO relations, which cannot be trained. The $W$ and $b$ are trainable parameters.

To obtain the important GO terms or important genes in the network, we use the Deep-LIFT scheme as implemented in P-NET [15, 26]. The DeepLIFT solution calculates important scores via backpropagation and can find the example-specific explanations when given an example and output. In our case, the calculated important scores can show how the input genes affect the disease through the GO function network. Given a certain sample $s$, $n_1, \ldots, n_l$ to be the number of nodes in the certain layer, and the specific target $y$, DeepLIFT calculates an importance score $C_i^{l,s}$ for each node $i$ based on the difference in the target activation $y - y_0$

fed by the certain sample $s$. The difference in target activation equals the sum of all node scores when fed by the certain sample $s$. That is,

$$\Delta y = y - y_0 = \sum_{i=1}^{n_l} C_i^{l,s} \tag{1}$$

We calculate the sample-level importance of all nodes in all layers using the 'Rescale rule' in DeepLIFT and calculate the total node-level importance score $C_i^l$ by aggregating the sample-level importance score over all the $n_s$ testing set samples.

$$C_i^l = |\sum_{i=1}^{n_s} C_i^{l,s}| \tag{2}$$

To reduce the bias introduced by over-annotation of certain nodes, we adjust the node important score by node degree $d_i^l$. The adjust node important score is calculated by:

$$adjusted\_C_i^l = \begin{cases} \dfrac{C_i^l}{d_i^l}, d_i^l > \mu + 5\sigma \\ \\ C_i^l \ (\text{otherwise}), \end{cases}$$

where $\mu$ is the mean of node degrees and $\sigma$ is the standard deviation of node degrees.

## Evaluation protocols

The evaluation protocols of the models were divided into two steps. In the first step, we randomly divided our dataset into 90% of the training and validation set and 10% of the testing set for hyperparameter tuning. We performed 10-fold cross-validation on the training-validation set to obtain the best hyperparameter settings for each model. In the second step, we performed a 10-fold cross-validation on the testing set to obtain the final evaluation metrics. The prediction performance was measured using accuracy, precision, recall, AUC, AUPRC, and F1score.

## Parameters settings

For GO informed model, we use hyperparameters grid search to find the best settings. The details of hyperparameters settings were: learning rate (0.01 / 0.005 / 0.001 / 0.0005 / 0.0001); regularizers (l1/l2/l1l2); reg_weight patterns (pattern1:[2,7,20,54,148,400](P-net default) / pattern2:[1,2,2,4,16,395](1/layer nodes number) / pattern3:[1,1,1,1,1,1] / pattern4:[1,2,4,8,16,32]). The loss function is binary crossentropy function and the optimization algorithm is Adam optimizer in GO informed model. To compare the disease predicting performance with GO informed model and the non-GO informed model, we utilized the AutoNN model as baseline [5]. AutoNN is a fully connected neural network with a certain hidden layer, and the number of nodes in each hidden layer is determined by the number of input layer nodes and layer number. The key distinction between GO-NN and AutoNN is that GO-NN utilizes gene-GO annotation information and the GO hierarchical structure to prune edges within the model, whereas AutoNN is a fully connected network without any biological information incorporated into the network structure. The details of hyperparameters settings of AutoNN were: hidden layer (1/2/3); drop rate (0/0.1); learning rate (0.01/0.001); and Adam optimizer. The number of learnable parameters of each model is shown in Table 2. AutoNN-h$x$ represents for AutoNN model with $x$ hidden layer.

Additionally, we compare the disease predicting performance with different machine learning models: support vector machine (SVM), random forest (RF), and logistic regression (LR).

**Table 2. Learnable parameters number in different neural network models.**

| GO-NN | AutoNN-h1 | AutoNN-h2 | AutoNN-h3 |
|---|---|---|---|
| 131,619 | 788,448,067 | 992,856,283 | 1,182,689,296 |

We use hyperparameters grid search to find the best settings. The details of hyperparameters settings were: the type of kernel (linear/polynomial) and the error term penalty (0.25/0.5/0.75/1.0/1.25/1.5/1.75/2.0) for the SVM; the splitting criterion (entropy/gini), the maximum tree depth (2/6/10), and the number of trees (10/50/100) for the RF; the penalty (l1/l2/elasticnet), solver (newton-cg/lbfgs/liblinear/sag/saga) and the inverse of regularization strength (0.25/0.5/0.75/1.0/1.25/1.5/1.75/2.0) for the logistic regression.

## Results

### Disease predicting performance comparison

To evaluate the effectiveness of the GO-informed model, we compared the disease predicting performance between the GO-informed model and non-GO-informed models. The classification results in T2D, liver cirrhosis dataset, inflammatory bowel disease, and colorectal cancer is shown in Table 3. We found that GO-NN has a better performance in the diabetes dataset (AUC = 0.778), inflammatory bowel disease dataset (F1score = 0.876, AUPRC = 0.979), and colorectal cancer dataset (F1score = 0.841, AUPRC = 0.937). RF has a better performance in

**Table 3. Classification result in four different datasets.**

| Dataset | Method | Accuracy | Precision | Reacll | F1Score | AUC | AUPRC |
|---|---|---|---|---|---|---|---|
| T2D | GO_NN | **0.708±0.055** | 0.647±0.041 | **0.958±0.042** | **0.772±0.040** | **0.778±0.051** | **0.776±0.048** |
| | AutoNN | 0.565±0.068 | 0.578±0.069 | 0.547±0.120 | 0.558±0.091 | 0.604±0.065 | 0.637±0.062 |
| | SVM | 0.581±0.018 | 0.555±0.010 | 0.926±0.026 | 0.694±0.015 | 0.495±0.023 | 0.552±0.010 |
| | RF | 0.681±0.036 | **0.698±0.033** | 0.668±0.062 | 0.682±0.042 | 0.733±0.022 | 0.751±0.031 |
| | LR | 0.511±0.056 | 0.522±0.051 | 0.600±0.071 | 0.557±0.053 | 0.545±0.058 | 0.521±0.033 |
| LC | GO_NN | 0.884±0.044 | 0.912±0.066 | 0.871±0.037 | 0.890±0.040 | 0.931±0.038 | 0.952±0.028 |
| | AutoNN | 0.791±0.059 | 0.815±0.069 | 0.788±0.060 | 0.800±0.056 | 0.862±0.062 | 0.868±0.064 |
| | SVM | 0.750±0.050 | 0.742±0.043 | 0.812±0.051 | 0.775±0.045 | 0.849±0.041 | 0.704±0.048 |
| | RF | **0.934±0.033** | **0.958±0.037** | **0.918±0.039** | **0.937±0.032** | **0.974±0.013** | **0.978±0.015** |
| | LR | 0.822±0.044 | 0.826±0.050 | 0.847±0.084 | 0.834±0.048 | 0.901±0.029 | 0.781±0.048 |
| IBD | GO_NN | **0.791±0.018** | 0.780±0.016 | **1.000±0.000** | **0.876±0.010** | **0.936±0.037** | **0.979±0.013** |
| | AutoNN | 0.781±0.086 | 0.809±0.089 | 0.917±0.093 | 0.853±0.060 | 0.821±0.057 | 0.910±0.069 |
| | SVM | 0.770±0.020 | **0.817±0.020** | 0.888±0.018 | 0.851±0.012 | 0.728±0.021 | 0.808±0.018 |
| | RF | 0.783±0.051 | 0.804±0.035 | 0.935±0.041 | 0.864±0.032 | 0.806±0.083 | 0.915±0.046 |
| | LR | 0.687±0.033 | 0.771±0.032 | 0.824±0.000 | 0.796±0.017 | 0.640±0.045 | 0.765±0.027 |
| CRC | GO_NN | 0.873±0.066 | 0.892±0.148 | **0.817±0.095** | **0.841±0.069** | **0.939±0.030** | **0.937±0.027** |
| | AutoNN | 0.787±0.065 | 0.888±0.139 | 0.550±0.150 | 0.665±0.122 | 0.874±0.060 | 0.849±0.075 |
| | SVM | **0.887±0.043** | **0.980±0.060** | 0.733±0.082 | 0.836±0.063 | 0.893±0.031 | 0.827±0.071 |
| | RF | 0.780±0.123 | 0.824±0.193 | 0.567±0.249 | 0.648±0.224 | 0.844±0.086 | 0.801±0.112 |
| | LR | 0.767±0.061 | 0.813±0.128 | 0.550±0.107 | 0.651±0.102 | 0.852±0.038 | 0.633±0.099 |

Comparison of the predicting performance of GO-informed NN (GO_NN), non-GO-informed NN (AutoNN, LaPierre et al.), and other machine learning models in different diseases. The mean and the standard deviation are recorded for different evaluation metrics after a 10-fold cross-validation. The best performances of each metric are highlighted in bold separately.

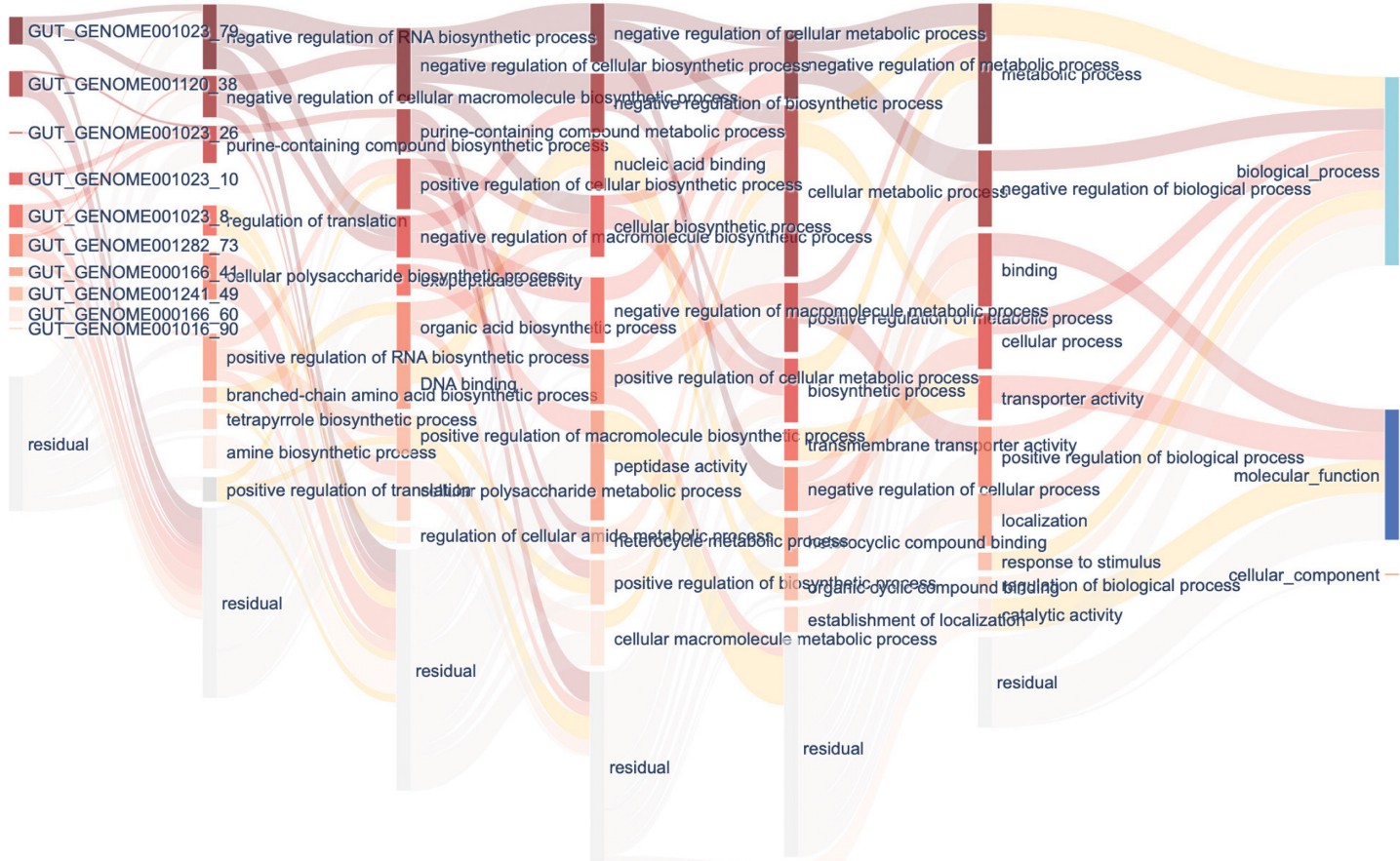

**Fig 3. Interpretation of GO informed neural network in diabetes.** The first layer shows the top 10 important genes. The following layers show the top 10 important GO terms in each level. The final layer shows the roots of each ontology. Nodes with darker colors have a larger important score. The transparent nodes represent the undisplayed nodes in each layer. Links with darker colors have larger edge weights.

the liver cirrhosis dataset (AUC = 0.974). In addition, the GO-informed model performs better than the non-GO-informed neural network model (AutoNN).

## GO informed neural network visualization

To understand how the microbes affect human diseases, we visualized the GO-informed neural network after training the diabetes dataset (Fig 3) and inflammatory bowel disease dataset (Fig 4). The first layer represents genes; the next layer represents GO terms in level 6, where genes are annotated; the next continues with level5 to level2 GO terms; the final layer represents the root GO terms: biological process, molecular function, and cellular component, which are directly connected with the outcome. We calculate the node's important score in the best fitting fold. We select the top 10 node important score genes in the input layer, and the top 10 nodes important score GO terms in each level except the last level with 3 GO terms. GO terms with an important score less than 1e-10 in each layer are not shown in the figure. Nodes with darker colors have a larger important score. The transparent nodes represent the undisplayed nodes in each layer. Links with darker colors have larger edge weights.

In diabetes classification, we detected 10 genes which exist in microbe species *Lachnospira* (GUT_GENOME 001023), *Bacteroides thetaiotaomicron* (GUT_GENO ME001120), *Prevotella*

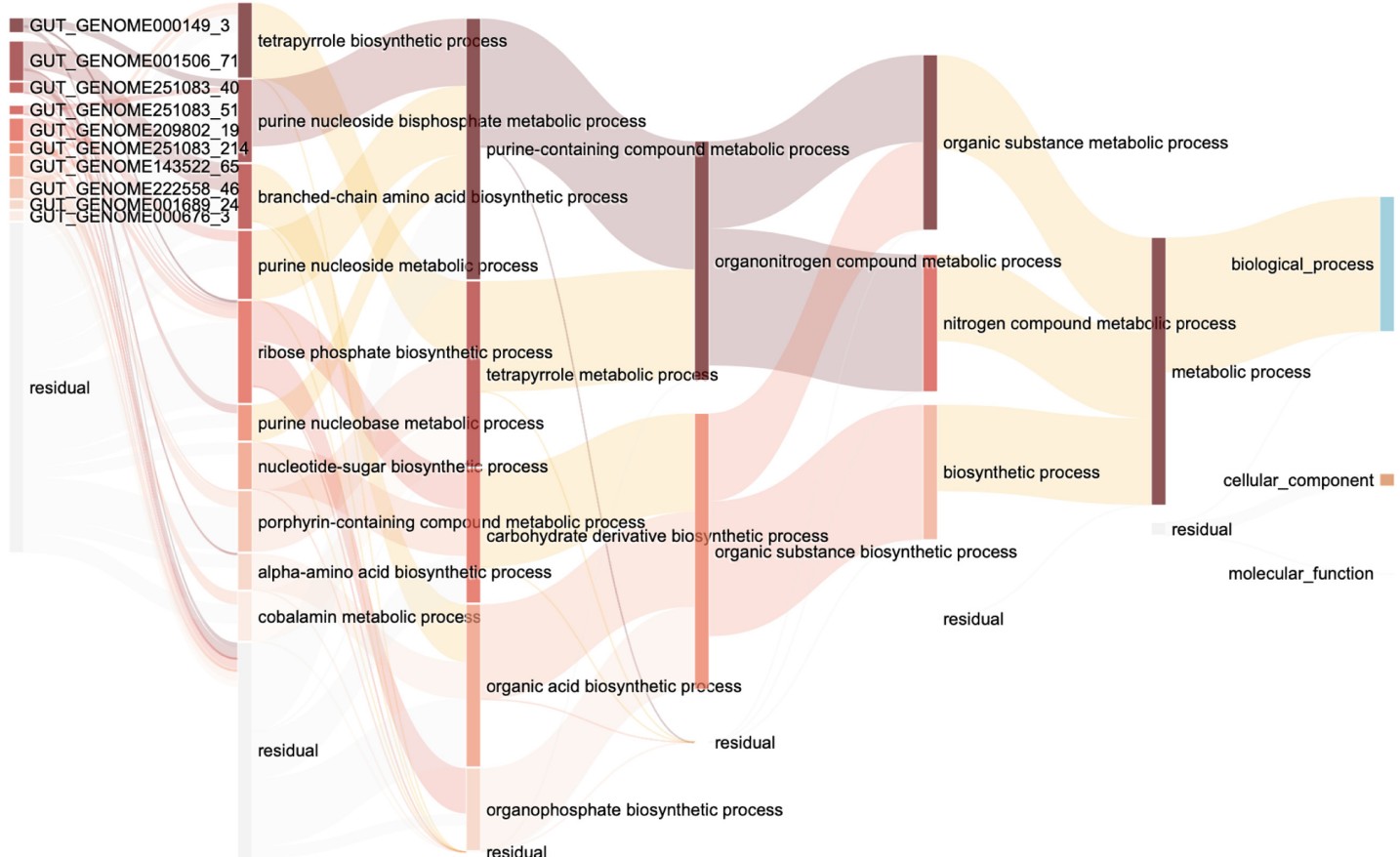

**Fig 4. Interpretation of GO informed neural network in inflammatory bowel disease.** The first layer shows the top 10 important genes. The following layers show the top 10 important GO terms in each level except the GO terms with an important score less than 1e-10. The final layer shows the roots of each ontology. Nodes with darker colors have a larger important score. The transparent nodes represent the undisplayed nodes in each layer. Links with darker colors have larger edge weights.

stercorea (GUT_GENOME001282), *Acetatifactor* (GUT_GE NOME000166), *Eubacterium_F* (GUT_GENOME001241), and *Agathobaculum butyriciproducens* (GUT_GENOME 001016) have important roles. In these species, *Lachnospira*, *Bacteroides thetaiotaomicron*, and *Prevotella stercorea* were reported to be associated with diabetes [27–29]. In these species, *Lachnospira* contains four important genes that contribute to negative regulation of RNA biosynthetic process, negative regulation of cellular macromolecule biosynthetic process, and purine-containing compound biosynthetic process.

In inflammatory bowel disease classification, we detected 10 genes which exist in microbe species *Dorea longicatena* (GUT_GENOME000149), *Roseburia sp003470905* (GUT_GENOME001506), *Gemmiger qucibialis* (GUT_GENOME251083), *Faecalibacterium sp900539885* (GUT_GENOME209802), *Angelakisella sp004557855* (GUT_GENOME222558), *Anaerobutyricum hallii* (GUT_GENOME001689), and *Blautia massiliensis* (GU T_GENOME000676) have important roles. In these species, *Dorea longicatena* was reported to be associated with inflammatory bowel disease [30]. In these species, *Gemmiger qucibialis* contains three important genes that contribute to the purine nucleoside bisphosphate metabolic process, ribose phosphate biosynthetic process, and purine nucleobase metabolic process. Most of the selected gene ontology terms with a high important score in inflammatory bowel disease come from the biological process.

### The relationship between the input gene number and the GO_NN model performance

Determining the input gene number of GO informed model is a crucial task. Selecting a large geneset will increase the number of parameters in the model. Simultaneously, genes that existed in a few samples which are considered as noises may misestimate as important genes. On the other hand, a small geneset will exclude the genes associated with the disease. To find the effect of input gene number in prediction result, we prepare a gene set where the genes exist more than 1%, 5%, and 10% of UHGG samples as commonly existing genes, which we named as Dataset-large, Dataset-median, Dataset-small separately. The number of genes selected in each gene set is 22,927, 7,663, and 3,451, separately. We compared the predicting performance of GO-informed NN in different gene sets in diabetes and liver cirrhosis, which is shown in Table 4. The result shows that the GO-informed model has higher performance (F1score, AUC, and AUPRC) in larger geneset in both diseases. Noted that there is a small performance gap between large geneset and medium geneset, which indicated that further increasing the input gene number has a little improvement in predicting performance.

### Precision-recall curves comparison between GO_NN and RF

We noticed that the performance of random forest is competitive by comparing with other machine learning models in diseases such as liver cirrhosis. Therefore, we performed a further analysis by comparing GO_NN and RF results. The precision-recall curves comparison of GO_NN and RF in different diseases were shown in Fig 5. From the precision-recall curves, there are less difference between the performance of two models in non-gastrointestinal disease including diabetes and liver cirrhosis datasets (Fig 5A and 5B). and larger difference in gastrointestinal disease including inflammatory bowel disease and colorectal cancer (Fig 5C and 5D). The overall performance of diabetes is lower than the other diseases, showing that machine learning models have difficulty in predicting diabetes. Limited information on the gene features results in difficulty in improving the performance, which is consistent with the previous study [1, 5].

**Table 4. Classification result in different genesets.**

| Dataset | Accuracy | Precision | Recall | F1score | AUC | AUPRC |
|---|---|---|---|---|---|---|
| T2D-Large | **0.708±0.055** | **0.647±0.041** | **0.958±0.042** | **0.772±0.040** | **0.778±0.051** | **0.776±0.048** |
| T2D-Medium | 0.667±0.067 | 0.631±0.054 | 0.858±0.079 | 0.726±0.055 | 0.708±0.044 | 0.708±0.060 |
| T2D-Small | 0.562±0.082 | 0.557±0.066 | 0.795±0.063 | 0.652±0.053 | 0.639±0.061 | 0.709±0.054 |
| LC-Large | **0.884±0.044** | **0.912±0.066** | 0.871±0.037 | **0.890±0.040** | **0.931±0.038** | **0.952±0.028** |
| LC-Medium | **0.884±0.021** | 0.905±0.037 | **0.876±0.019** | **0.890±0.018** | 0.924±0.028 | 0.943±0.028 |
| LC-Small | 0.794±0.051 | 0.821±0.076 | 0.794±0.042 | 0.805±0.039 | 0.857±0.047 | 0.888±0.047 |
| IBD-Large | **0.791±0.018** | 0.780±0.016 | **1.000±0.000** | **0.876±0.010** | **0.936±0.037** | **0.979±0.013** |
| IBD-Medium | 0.778±0.038 | **0.794±0.030** | 0.947±0.043 | 0.863±0.024 | 0.807±0.042 | 0.931±0.019 |
| IBD-Small | 0.770±0.054 | 0.792±0.034 | 0.935±0.052 | 0.857±0.035 | 0.795±0.070 | 0.927±0.029 |
| CRC-Large | **0.873±0.066** | **0.892±0.148** | **0.817±0.095** | **0.841±0.069** | **0.939±0.030** | **0.937±0.027** |
| CRC-Medium | 0.760±0.072 | 0.714±0.132 | 0.733±0.225 | 0.695±0.124 | 0.907±0.050 | 0.884±0.057 |
| CRC-Small | 0.700±0.085 | 0.650±0.158 | 0.717±0.273 | 0.637±0.128 | 0.887±0.068 | 0.844±0.100 |

Comparison of the predicting performance of GO informed NN in different gene sets in diabetes (T2D), liver cirrhosis (LC), inflammatory bowel disease (IBD), and colorectal cancer (CRC). The mean and the standard deviation are recorded for different evaluation metrics after a 10-fold cross-validation. The best performances of each metric are highlighted in bold separately.

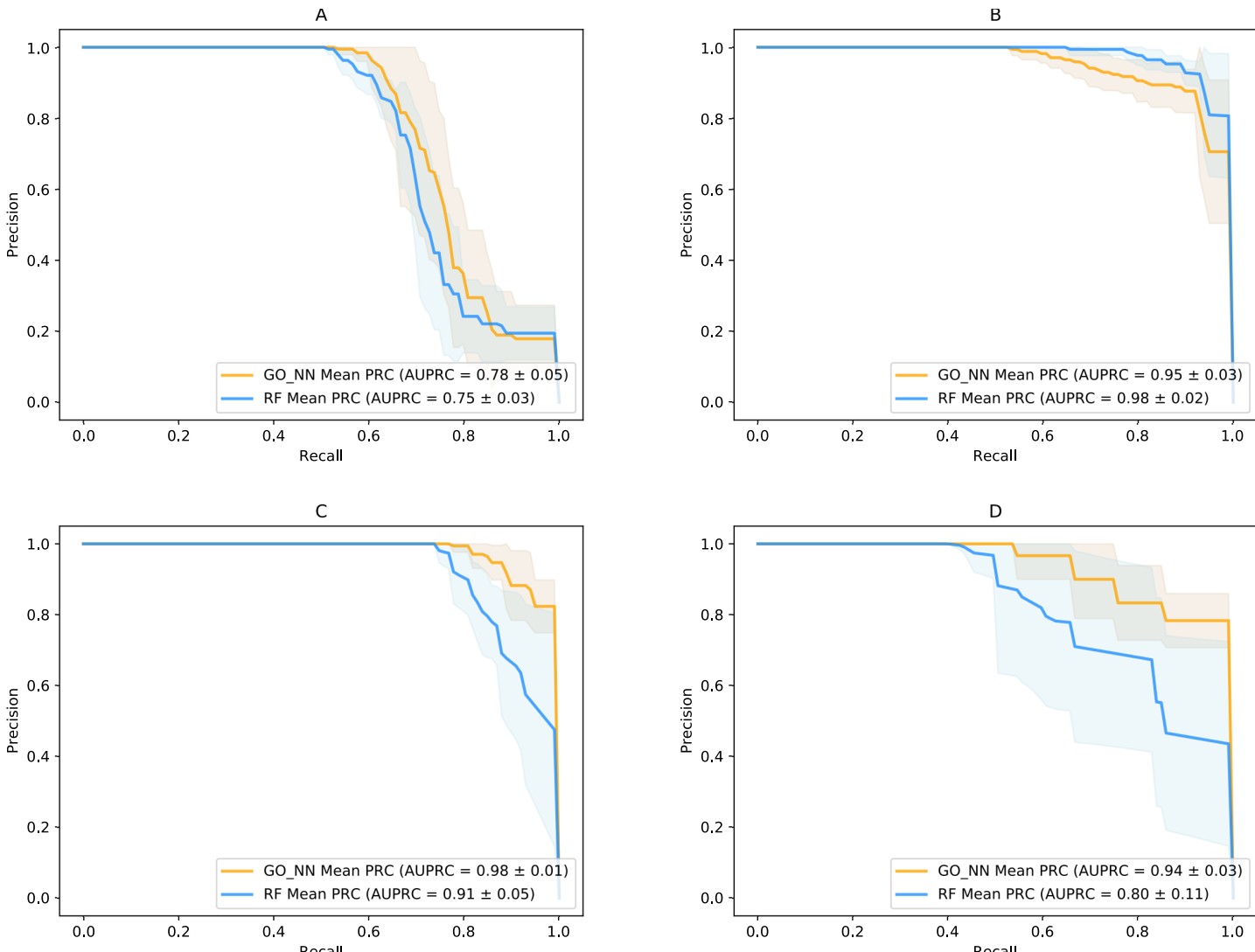

**Fig 5. Precision-recall curves comparison of GO_NN and RF in different diseases.** The solid line and the shadow represent the mean and standard deviation of 10-fold cross-validation results. **A** T2D precision-recall curves. **B** LC precision-recall curves. **C** IBD precision-recall curves. **D** CRC precision-recall curves.

## Discussion

The disease predicting result shows the effectiveness of the GO-informed neural network in predicting different diseases, especially gastrointestinal diseases. GO-NN gives a competitive result in different datasets and shows the importance of candidate genes and their functions. In addition, GO-informed neural network reduces the number of parameters for learning by utilizing genes GO annotation information and GO hierarchical structure. Compared with the non-GO-informed neural network, GO-informed neural network has fewer learnable parameters and overall disease prediction performance.

Furthermore, the visualization of GO informed model explains microbe functionality by integrating metagenome species, metagenome genes, and GO information. The network provides the functionality explanation of metagenome genes, which has the potential to discover novel species and functions that affect the disease. Specifically, GO informed model observes

the important species not reported in previous research in both diabetes and liver cirrhosis datasets. These species have important functional roles in the disease, which cannot be discovered by a non-GO informed model.

Whereas GO informed model provides better performance than the non-GO-informed model, there are some issues with improving the performance of the GO-informed model. Firstly, we noted that the gene number affects the performance of GO informed model. Using a larger geneset helps improve the performance of GO informed model in both diseases. In addition, the sample size is still much smaller than the feature size. The GO-informed model performance may improve by using more qualified samples. Moreover, using heterogeneous data by combining GO with other biological priors, such as KEGG, may further guide model development and functional evaluation.

## Conclusion

In conclusion, we propose to utilize a GO-informed neural network to discover the microbe functionality in human diseases, which existing models cannot obtain. The GO-informed neural network model has effectiveness in disease prediction in diabetes and liver cirrhosis datasets. Our model discovered the important microbiome function, genes, and microbe species by calculating the important score of each gene and GO term in the network. We visualized the network's important genes and GO terms and provided insights into microbe contribution in functional aspects, which has the potential for clinical translation in disease-specified microbe-involved functions.

## Author Contributions

**Data curation:** Yunjie Liu.

**Methodology:** Yunjie Liu.

**Supervision:** Yao-zhong Zhang, Seiya Imoto.

**Visualization:** Yunjie Liu.

**Writing – original draft:** Yunjie Liu.

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
