## [Decision Letter · Decision Letter 0]

21 Mar 2023

PONE-D-23-05707Microbial Gene Ontology informed deep neural network for microbe functionality discovery in human diseasesPLOS ONE

Dear Dr. Liu,

Thank you for submitting your manuscript to PLOS ONE. After careful consideration, we feel that it has merit but does not fully meet PLOS ONE’s publication criteria as it currently stands. Therefore, we invite you to submit a revised version of the manuscript that addresses the points raised during the review process.

Please revise by considering both reviewers comments thoroughly. Please pay particular attention to comments on comparison with other state-of-the-art methods and biological interpretation of results. Also please make your tool available on github or other code repositories with detailed documentation to benefit the user community.

We look forward to receiving your revised manuscript.

Kind regards,

Yanbin Yin

Academic Editor

PLOS ONE

Journal Requirements:

3. We note that you have stated that you will provide repository information for your data at acceptance. Should your manuscript be accepted for publication, we will hold it until you provide the relevant accession numbers or DOIs necessary to access your data. If you wish to make changes to your Data Availability statement, please describe these changes in your cover letter and we will update your Data Availability statement to reflect the information you provide

Reviewers' comments:

Reviewer's Responses to Questions

**Comments to the Author**

1. Is the manuscript technically sound, and do the data support the conclusions?

Reviewer #1: Partly

Reviewer #2: Partly

2. Has the statistical analysis been performed appropriately and rigorously? 

Reviewer #1: No

Reviewer #2: Yes

3. Have the authors made all data underlying the findings in their manuscript fully available?

Reviewer #1: No

Reviewer #2: No

4. Is the manuscript presented in an intelligible fashion and written in standard English?

Reviewer #1: No

Reviewer #2: Yes

5. Review Comments to the Author

Reviewer #1: This article presents an algorithm for inferring gene function based on gene abundance, which bears a striking resemblance to the P-Net algorithm published in Nature in 2021. While the algorithm is innovative, the lack of many methodological details in the article raises doubts about its value. Here are a few questions I am concerned about.

1. In the left side of Figure 1, the process for obtaining the abundance of each gene is shown. Therefore, the text "Metagenome sequence" should be corrected to "sequencing reads" instead of "genome"."

2. How does the author obtain the GO annotation for each UHGG gene? The author only mentions “The GO annotation information is obtained by referring to the UniProt database”. Using blastp? What parameters are used?

3. The author claims “We prepare a gene set where the genes exist in more than 1% of 67 UHGG samples as commonly existing genes”. What is the detail? How many reads or TPM are regards as “exist in a sample”. As we know, some low abundance genes may be due to the sequencing error.

4. Figure 1 shows the depth of the GO-informed neural network is 4 layers (exclude input layer). However, in line 95-97, the authors mention “The second layer to the seventh layer”. So, how many layers are used? The author should correct the error in the figure 1.

5. Formula: y = f[(M ∗W)T xlayer_i +b] (i = 6, 5, 4, 3, 2, 1) shows how the connection between layers. M is the mask matrix is dependent on the GO annotation of the genes, which can’t be trained. The trainable parameters should be W and b. The author should add more sentences to illustrate the GO network.

6. What is the y0? How did the author prepare the target?

7. How did the network train? Which Loss function, optimization algorithm? The author should provide a figure to show the training process (Loss vs epoch).

8. The author should make the source code to open access.

Reviewer #2: I have reviewed your paper and found it to be interesting and informative. However, there are several issues that need to be addressed before the paper can be accepted for publication.

1. The arrangement of the figure caption and table caption is in a mess. Please rearrange them.

2. Regarding your evaluation protocol, it would be helpful to know how you divided the training+validation and testing sets. Did you do this randomly or intentionally? If you swap the training+validation and testing, what would be the resulting performance?

3. Why did you use a 9:1 split ratio? Additionally, besides the evaluation metrics provided, could you provide a biological interpretation to help illustrate the reliability of your model?

4. For the SVM kernel, did you try using the RBF kernel? If so, how did its performance compare to your method? Also, have you tried using XGBoost? As it is known to achieve state-of-the-art performance compared to other machine learning methods.

5. Could you please provide information on your beta and lambda grid search settings for your elastic net model?

6. AutoNN achieves worse performance than other methods. Have you tried combining GO with other machine learning or statistical learning methods that you benchmarked? It may not be necessary to use a deep learning model like AutoNN.

7. The precision of 0.522 to 0.698 are relatively low, indicating a high number of false positives in the classification of T2D. Could you explain why this may be the case?

8. I recommend using DeepLift to calculate the feature importance of your model. I want to see the shap values comparison from the feature of your AutoNN. Also, please apply SHAP method to explain those machine learning and statistical learning methods that you benchmarked.

9. Could you please provide the data URL in your Github? I was unable to locate it, and why did you only provide T2D and LC data loader codes? Where are the IBD and CRC data?

10. Please included a comparison with other state-of-the-art methods that predict human disease from microbiota, such as those presented in "Multimodal deep learning applied to classify healthy and disease states of human microbiome" (https://www.nature.com/articles/s41598-022-04773-3) and "DeepMicro: deep representation learning for disease prediction based on microbiome data" (https://www.nature.com/articles/s41598-020-63159-5). It would provide a more comprehensive understanding of the performance of the proposed method in comparison to other relevant approaches.

Please address these issues in a revised version of your manuscript. I look forward to reviewing the updated version.

6. PLOS authors have the option to publish the peer review history of their article (what does this mean?). If published, this will include your full peer review and any attached files.

Reviewer #1: No

Reviewer #2: No

---

## [Author Response · Author response to Decision Letter 0]

21 May 2023

Responds to the reviewers’ comments 

We sincerely thank all reviewers on the valuable comments on this manuscript.

Reviewer #1: This article presents an algorithm for inferring gene function based on gene abundance, which bears a striking resemblance to the P-Net algorithm published in Nature in 2021. While the algorithm is innovative, the lack of many methodological details in the article raises doubts about its value. Here are a few questions I am concerned about.

1. In the left side of Figure 1, the process for obtaining the abundance of each gene is shown. Therefore, the text "Metagenome sequence" should be corrected to "sequencing reads" instead of "genome"."

[Liu response]: Thank you for pointing this out. The text has been corrected in the Figure 1.

2. How does the author obtain the GO annotation for each UHGG gene? The author only mentions “The GO annotation information is obtained by referring to the UniProt database”. Using blastp? What parameters are used?

[Liu response]: Thank you for pointing this out. First, we obtain the protein annotation from each gene provided by the UHGG and find out the gene-protein mapping. Then, we search the Gene Ontology annotation of each protein from the Uniprot database to find the out the protein-GO mapping. Finally, by merging the gene-protein mapping and protein-GO mapping, we obtain the GO annotation for each UHGG gene.

3. The author claims “We prepare a gene set where the genes exist in more than 1% of 67 UHGG samples as commonly existing genes”. What is the detail? How many reads or TPM are regards as “exist in a sample”. As we know, some low abundance genes may be due to the sequencing error.

[Liu response]: Thank you for pointing this out. The number of genes selected in the gene set where the genes exist in more than 1% of UHGG samples is 22,927. Considering we have filtered the gene set by using commonly existing genes, genes of TPM>0 were regarded as existing in a sample. 

4. Figure 1 shows the depth of the GO-informed neural network is 4 layers (exclude input layer). However, in line 95-97, the authors mention “The second layer to the seventh layer”. So, how many layers are used? The author should correct the error in the figure 1.

[Liu response]: Thank you for pointing this out. The complete GO-informed neural network includes 1 input layer, 6 GO hierarchical structure layers form GO level 6 to GO level 1, and 1 predictive layer. We have corrected the GO hierarchical structure layers part in the Figure 1.

5. Formula: y = f[(M ?W)T xlayer_i +b] (i = 6, 5, 4, 3, 2, 1) shows how the connection between layers. M is the mask matrix is dependent on the GO annotation of the genes, which can’t be trained. The trainable parameters should be W and b. The author should add more sentences to illustrate the GO network.

[Liu response]: Thank you for your suggestion. We have added the description of GO network in the section "GO network informed DNN construction".

6. What is the y0? How did the author prepare the target?

[Liu response]: Thank you for pointing this out. The DeepLIFT-based important scores were calculated to visualize the important genes and GO terms in a specific disease. Therefore, we use the whole dataset in a specific disease as the target to calculate the important scores. In this case, y0 will be the disease label of whole dataset in a specific disease.

7. How did the network train? Which Loss function, optimization algorithm? The author should provide a figure to show the training process (Loss vs epoch).

[Liu response]: Thank you for pointing this out. The loss function is binary crossentropy function and the optimization algorithm is Adam optimizer. We have added the description in the "Parameters settings" section. The training curves for are shown in the Supplemental Figure 1. ~ Supplemental Figure 4.

Supplemental Figure 1. The training curves in T2D. A: training loss and validation loss vs epoch. B: F1-score vs epoch in training set in each layer. C: F1-score vs epoch in validation set in each layer.

Supplemental Figure 2. The training curves in LC. A: training loss and validation loss vs epoch. B: F1-score vs epoch in training set in each layer. C: F1-score vs epoch in validation set in each layer.

Supplemental Figure 3. The training curves in IBD. A: training loss and validation loss vs epoch. B: F1-score vs epoch in training set in each layer. C: F1-score vs epoch in validation set in each layer.

Supplemental Figure 4. The training curves in CRC. A: training loss and validation loss vs epoch. B: F1-score vs epoch in training set in each layer. C: F1-score vs epoch in validation set in each layer.

8. The author should make the source code to open access.

[Liu response]: Thank you for pointing this out. We have made the source code to open access.

Reviewer #2: I have reviewed your paper and found it to be interesting and informative. However, there are several issues that need to be addressed before the paper can be accepted for publication.

1. The arrangement of the figure caption and table caption is in a mess. Please rearrange them.

[Liu response]: Thank you for pointing this out. We have adjusted the figure caption and table caption.

2. Regarding your evaluation protocol, it would be helpful to know how you divided the training+validation and testing sets. Did you do this randomly or intentionally? If you swap the training+validation and testing, what would be the resulting performance?

[Liu response]: Thank you for pointing this out. The training+validation and testing sets are split randomly. Since the sample size is small and we use 9:1 split ratio to split training+validation and testing, if we swap the training+validation and testing, the samples to be learned will be very small and the model will underfitted in this case.

3. Why did you use a 9:1 split ratio? Additionally, besides the evaluation metrics provided, could you provide a biological interpretation to help illustrate the reliability of your model?

[Liu response]: Thank you for pointing this out. Considering the sample size is small, we use a 9:1 split ratio to make the model to learn well. In the "GO informed neural network visualization" section, we have provided the example biological interpretation in T2D and IBD diseases.

4. For the SVM kernel, did you try using the RBF kernel? If so, how did its performance compare to your method? Also, have you tried using XGBoost? As it is known to achieve state-of-the-art performance compared to other machine learning methods.

[Liu response]: Thank you for pointing this out. In SVM kernel, we test only linear and polynomial kernel. For XGBoost, RF and XGBoost are both decision tree algorithms and RF have been shown good performance in previous research about using metagenome sequence-based feature to do the disease prediction. Therefore, we only test the performance of RF in our study.

5. Could you please provide information on your beta and lambda grid search settings for your elastic net model?

[Liu response]: Thank you for pointing this out. We did not perform the grid search on beta and lambda for the elastic net penalty in the logistic regression model. We use the default settings of l1 ratio=0.5. Instead, we test l1 and l2 penalty as the special cases of the elastic net model. 

6. AutoNN achieves worse performance than other methods. Have you tried combining GO with other machine learning or statistical learning methods that you benchmarked? It may not be necessary to use a deep learning model like AutoNN.

[Liu response]: Thank you for pointing this out. Our GO model did not combine with AutoNN or any other machine learning models. AutoNN was used as a representative NN model to compare the performance of our model and other machine learning models.

7. The precision of 0.522 to 0.698 are relatively low, indicating a high number of false positives in the classification of T2D. Could you explain why this may be the case?

[Liu response]: Thank you for pointing this out. In T2D dataset, it is a difficult dataset from the performance of different models (precision from 0.522 to 0.698). Combining with the training curve in Supplemental Figure. 1, we find the model overfitted in this dataset and it cannot generate well in the validation set. One main reason of this case is that T2D is a complex disease and the disease is not only related to the gut microbiome, but also related to other factors such as human genetics. 

8. I recommend using DeepLift to calculate the feature importance of your model. I want to see the shap values comparison from the feature of your AutoNN. Also, please apply SHAP method to explain those machine learning and statistical learning methods that you benchmarked.

[Liu response]: Thank you for pointing this out. To compare the DeepLift importance score in our model and other model's SHAP value, we calculate the SHAP value of AutoNN, RF, and SVM in T2D and IBD dataset separately using SHAP (Lundberg, S. M., & Lee, S. I. (2017). A unified approach to interpreting model predictions. Advances in neural information processing systems, 30.). We plot top 10 SHAP value in different models (see Supplemental Figure. 5 and Supplemental Figure. 6).

Supplemental Figure 5. T2D top 10 SHAP value in different models. A: T2D top 10 SHAP value in AutoNN. B: T2D top 10 SHAP value in RF. C: T2D top 10 SHAP value in SVM.

Supplemental Figure 6. IBD top 10 SHAP value in different models. A: IBD top 10 SHAP value in AutoNN. B: IBD top 10 SHAP value in RF. C: IBD top 10 SHAP value in SVM.

From the SHAP value and DeepLift results (Figure 3. and Figure 4), we find that GO-NN DeepLift-important-score based important genes are different by comparing with other models. This is because our important genes are related to the GO network while other models do not have this constraint. Moreover, our model not only can find the important genes, but also can find the important GO terms and the relation between genes and GO terms, which can not be visualized in SHAP value in other models.

9. Could you please provide the data URL in your Github? I was unable to locate it, and why did you only provide T2D and LC data loader codes? Where are the IBD and CRC data?

[Liu response]: Thank you for pointing this out. We have updated the Github page and add the data URL. We also provide IBD and CRC data loader codes and the data. 

10. Please included a comparison with other state-of-the-art methods that predict human disease from microbiota, such as those presented in "Multimodal deep learning applied to classify healthy and disease states of human microbiome" (https://www.nature.com/articles/s41598-022-04773-3) and "DeepMicro: deep representation learning for disease prediction based on microbiome data" (https://www.nature.com/articles/s41598-020-63159-5). It would provide a more comprehensive understanding of the performance of the proposed method in comparison to other relevant approaches.

[Liu response]: While we appreciate the suggestions, we respectfully disagree for the following reasons. First, in these models, the feature they prepared is different with ours. In "Multimodal deep learning applied to classify healthy and disease states of human microbiome" (https://www.nature.com/articles/s41598-022-04773-3), they use bacterial proportion, metabolic function abundance, and genome-level abundance; In "DeepMicro: deep representation learning for disease prediction based on microbiome data" (https://www.nature.com/articles/s41598-020-63159-5), they use strain-level marker profile and species-level relative abundance profile. Second, these models are not designed for microbe functionality discovery, which is a major objective of our model.

---

## [Decision Letter · Decision Letter 1]

4 Jun 2023

PONE-D-23-05707R1Microbial Gene Ontology informed deep neural network for microbe functionality discovery in human diseasesPLOS ONE

Dear Dr. Liu,

Thank you for submitting your manuscript to PLOS ONE. After careful consideration, we feel that it has merit but does not fully meet PLOS ONE’s publication criteria as it currently stands. Therefore, we invite you to submit a revised version of the manuscript that addresses the points raised during the review process.

We look forward to receiving your revised manuscript.

Kind regards,

Yanbin Yin

Academic Editor

PLOS ONE

Journal Requirements:

Reviewers' comments:

Reviewer's Responses to Questions

**Comments to the Author**

1. If the authors have adequately addressed your comments raised in a previous round of review and you feel that this manuscript is now acceptable for publication, you may indicate that here to bypass the “Comments to the Author” section, enter your conflict of interest statement in the “Confidential to Editor” section, and submit your "Accept" recommendation.

Reviewer #1: All comments have been addressed

Reviewer #2: (No Response)

2. Is the manuscript technically sound, and do the data support the conclusions?

Reviewer #1: Yes

Reviewer #2: Partly

3. Has the statistical analysis been performed appropriately and rigorously? 

Reviewer #1: Yes

Reviewer #2: Yes

4. Have the authors made all data underlying the findings in their manuscript fully available?

Reviewer #1: Yes

Reviewer #2: Yes

5. Is the manuscript presented in an intelligible fashion and written in standard English?

Reviewer #1: Yes

Reviewer #2: Yes

6. Review Comments to the Author

Reviewer #1: (No Response)

Reviewer #2: Thank you for addressing my previous comments. I appreciate your efforts in resolving the majority of my concerns. However, there are a couple of points that still need further addressing:

Regarding my 4th question, I recommended employing SVM with an RBF kernel, given it frequently demonstrated superiority over its linear and polynomial counterparts. Regrettably, I remain unconvinced by your rationale for not adopting this approach. Similarly, XGBoost is widespread use and notable effectiveness across numerous studies, often exceeding Random Forests (RF), make it a worthwhile comparator to your method. However, your explanation for its exclusion lacks persuasive power. Both SVM with an RBF kernel and XGBoost are relatively simple to implement, and I would strongly encourage you to contemplate these comparisons.

Moving on to my 6th question, it appears that your model built from Gene Ontology (GO) and Neural Networks (NN) primarily benefits from the GO input, rather than the deep learning methodology or the NN model itself. In your comparison result (AUC in Table 3) among AutoNN (a representative of NN), SVM, RF, and Logistic Regression (LR), AutoNN does not emerge as the top performer, except for within your GO-informed model. Thus, I recommend you design a GO-informed Random Forest or a GO-informed SVM. If it is time-consuming, alternatively, you could simply acknowledge that your model derives more benefit from the GO input rather than the deep learning process.

Finally, concerning your supplementary figures, it would be more consistent and visually appealing to place the captions below the images, rather than above. Please make this amendment accordingly.

7. PLOS authors have the option to publish the peer review history of their article (what does this mean?). If published, this will include your full peer review and any attached files.

Reviewer #1: No

Reviewer #2: No

---

## [Author Response · Author response to Decision Letter 1]

21 Jul 2023

Responds to the reviewers’ comments 

We sincerely thank all reviewers on the valuable comments on this manuscript.

Reviewer #1: (No Response)

Reviewer #2: 

1. Regarding my 4th question, I recommended employing SVM with an RBF kernel, given it frequently demonstrated superiority over its linear and polynomial counterparts. Regrettably, I remain unconvinced by your rationale for not adopting this approach. Similarly, XGBoost is widespread use and notable effectiveness across numerous studies, often exceeding Random Forests (RF), make it a worthwhile comparator to your method. However, your explanation for its exclusion lacks persuasive power. Both SVM with an RBF kernel and XGBoost are relatively simple to implement, and I would strongly encourage you to contemplate these comparisons.

[Liu response]: Thank you for pointing this out. In our experiments, we followed the same procedure and data split as described in the "Evaluation protocols" section to test the performance of SVM with an RBF kernel and XGBoost. We utilized Scikit-learn 1.2.2 for implementing SVM with an RBF kernel, and XGBoost 1.5.2 for training and evaluating the XGBoost model. These versions of the libraries were used to ensure consistency and reproducibility in our experiments.

In the hyperparameter tuning step, we utilized commonly used settings to search for the best hyperparameter configurations. For SVM with an RBF kernel, we considered the following details of the hyperparameter settings: regularization parameter C (0.25, 0.5, 0.75, 1.0, 1.25, 1.5, 1.75, 2.0 (consistent with SVM linear/poly kernel)) and kernel coefficient gamma set to 'scale'. Regarding XGBoost, the hyperparameter settings were as follows: learning rate eta (0.001, 0.01, 0.1), maximum depth of a tree max_depth (3, 6, 10 (consistent with RF)), 'gbtree' booster, and 300 boosting rounds.

In the testing step, we conducted a 10-fold cross-validation using the best hyperparameter settings selected based on the highest F1Score, which is consistent with the approach used for other models. The final results are presented in Supplemental Table 1. We observed that GO-NN demonstrated the best performance in the T2D, IBD, and CRC datasets, achieving the highest F1Score, AUC, and AUPRC. Comparing SVM with linear/poly kernel to SVM with RBF kernel, we found that SVM with RBF kernel yielded a higher F1Score than SVM with linear/poly kernel in the LC dataset but a lower F1Score in the T2D, IBD, and CRC datasets. Regarding the comparison between XGBoost and RF, XGBoost outperformed RF in the IBD dataset, but had a lower F1Score than SVM with linear/poly kernel in the T2D, LC, and CRC datasets. While SVM with RBF kernel and XGBoost demonstrated better performance than SVM with linear/poly kernel and RF in the LC and IBD datasets, GO-NN still exhibited superior performance compared to SVM with RBF kernel and XGBoost in all evaluated datasets.

Supplemental Table 1. Comparison of the classification result in four different datasets

Dataset Method Accuracy Precision Recall F1Score AUC AUPRC

T2D SVM-RBF 0.568±0.032 0.576±0.031 0.595±0.067 0.584±0.043 0.626±0.021 0.552±0.018

 SVM 0.581±0.018 0.555±0.010 0.926±0.026 0.694±0.015 0.495±0.023 0.552±0.010

 XGBoost 0.622±0.065 0.648±0.070 0.579±0.100 0.608±0.077 0.623±0.065 0.595±0.050

 RF 0.681±0.036 0.698±0.033 0.668±0.062 0.682±0.042 0.733±0.022 0.751±0.031

 GO_NN 0.708±0.055 0.647±0.041 0.958±0.042 0.772±0.040 0.778±0.051 0.776±0.048

LC SVM-RBF 0.844±0.031 0.846±0.028 0.865±0.065 0.854±0.033 0.921±0.018 0.803±0.029

 SVM 0.750±0.050 0.742±0.043 0.812±0.051 0.775±0.045 0.849±0.041 0.704±0.048

 XGBoost 0.828±0.066 0.918±0.057 0.741±0.092 0.818±0.076 0.834±0.064 0.822±0.071

 RF 0.934±0.033 0.958±0.037 0.918±0.039 0.937±0.032 0.974±0.013 0.978±0.015

 GO_NN 0.884±0.044 0.912±0.066 0.871±0.037 0.890±0.040 0.931±0.038 0.952±0.028

IBD SVM-RBF 0.738±0.132 0.738±0.132 1.000±0.000 0.842±0.089 0.500±0.000 0.738±0.132

 SVM 0.770±0.020 0.817±0.020 0.888±0.018 0.851±0.012 0.728±0.021 0.808±0.018

 XGBoost 0.809±0.078 0.835±0.056 0.929±0.090 0.876±0.055 0.698±0.110 0.829±0.056

 RF 0.783±0.051 0.804±0.035 0.935±0.041 0.864±0.032 0.806±0.083 0.915±0.046

 GO_NN 0.791±0.018 0.780±0.016 1.000±0.000 0.876±0.010 0.936±0.037 0.979±0.013

CRC SVM-RBF 0.653±0.027 0.800±0.400 0.133±0.067 0.229±0.114 0.872±0.055 0.480±0.040

 SVM 0.887±0.043 0.980±0.060 0.733±0.082 0.836±0.063 0.893±0.031 0.827±0.071

 XGBoost 0.640±0.104 0.594±0.136 0.450±0.150 0.495±0.120 0.608±0.098 0.491±0.081

 RF 0.780±0.123 0.824±0.193 0.567±0.249 0.648±0.224 0.844±0.086 0.801±0.112

 GO_NN 0.873±0.066 0.892±0.148 0.817±0.095 0.841±0.069 0.939±0.030 0.937±0.027

The mean and the standard deviation are recorded for different evaluation metrics after a 10-fold cross-validation. The best performances of each metric are highlighted in bold separately.

2. Moving on to my 6th question, it appears that your model built from Gene Ontology (GO) and Neural Networks (NN) primarily benefits from the GO input, rather than the deep learning methodology or the NN model itself. In your comparison result (AUC in Table 3) among AutoNN (a representative of NN), SVM, RF, and Logistic Regression (LR), AutoNN does not emerge as the top performer, except for within your GO-informed model. Thus, I recommend you design a GO-informed Random Forest or a GO-informed SVM. If it is time-consuming, alternatively, you could simply acknowledge that your model derives more benefit from the GO input rather than the deep learning process.

[Liu response]: Thank you for pointing this out and sorry for the confusing description. 

In our model, we utilize the gene quantification profile as the input. The model incorporates gene-GO annotation information to establish connections between genes and level 6 Gene Ontology (GO) terms. Additionally, we utilize the hierarchical structure of the GO terms to connect level 6 GO terms to level 1 GO terms. To ensure the network learning based on the biological information, we employ a mask matrix (M) to filter out edges that involve genes without GO annotation and GO terms without relationships between adjacent GO levels. It's important to note that the GO information used in GO-NN is at the model level and not at the input level.

On the other hand, in AutoNN, the input gene quantification profile is fully connected to the hidden layer, and the network lacks the biological information of GO. The key distinction between GO-NN and AutoNN is that GO-NN utilizes gene-GO annotation information and the GO hierarchical structure to prune edges within the model, whereas AutoNN is a fully connected network without any biological information incorporated into the network structure.

When comparing GO-NN to other machine learning models, we use the same gene quantification profile as the input. While it is possible to construct a GO profile using gene-GO annotation information and use it as the input for SVM or RF, the results would not be directly comparable to GO-NN due to the differences in input representation and the incorporation of GO hierarchical structure information.

3. Finally, concerning your supplementary figures, it would be more consistent and visually appealing to place the captions below the images, rather than above. Please make this amendment accordingly.

[Liu response]: Thank you for pointing this out. We have placed the captions below the images.

---

## [Editor Report · Decision Letter 2]

7 Aug 2023

Microbial Gene Ontology informed deep neural network for microbe functionality discovery in human diseases

PONE-D-23-05707R2

Dear Dr. Liu,

We’re pleased to inform you that your manuscript has been judged scientifically suitable for publication and will be formally accepted for publication once it meets all outstanding technical requirements.

Kind regards,

Yanbin Yin

Academic Editor

PLOS ONE
---

## [Editor Report · Acceptance letter]

10 Aug 2023

PONE-D-23-05707R2 

Microbial Gene Ontology informed deep neural network for microbe functionality discovery in human diseases 

Dear Dr. Liu:

I'm pleased to inform you that your manuscript has been deemed suitable for publication in PLOS ONE. Congratulations! Your manuscript is now with our production department. 

Kind regards, 

on behalf of

Dr. Yanbin Yin 

Academic Editor

PLOS ONE